# Receiving support for mental health problems from family and friends: Measurement and impact on mental health, relationships, and help-seeking

**Amy J. Morgan****[1]\*, Judith Wright[1], Andrew J. Mackinnon[1], Nicola J. Reavley[1], Alyssia Rossetto[1], Long Khanh-Dao Le[2], Anthony F. Jorm[1]**

1 Centre for Mental Health and Community Wellbeing, Melbourne School of Population and Global Health, The University of Melbourne, Parkville, Australia, 2 Health Economics Group, School of Public Health and Preventive Medicine, Monash University, Melbourne, Australia

\* ajmorgan@unimelb.edu.au

## Abstract

Mental health-related discrimination by friends, family members and intimate partners is especially common, particularly in the form of avoidance and dismissive reactions. In contrast, good quality initial support (mental health first aid) is thought to improve help-seeking and mental health outcomes, although there are few studies that have examined this. This study aimed to develop a measure of mental health support skills that could be self-reported by the recipient of help, and to use this measure to test whether better quality support was related to better outcomes. We recruited 1116 adults from Prolific who reported a recent mental health problem or crisis and were helped by someone close to them such as a family member or friend. Participants completed a recipient version of the Mental Health Support Scale, which asked about actions the other person took to support them, based on mental health first aid guidelines. They also completed measures assessing convergent and divergent validity and perceived impact of the support from their helper. Confirmatory factor analysis supported a scale with two factors of recommended and not recommended actions with acceptable psychometric properties. Receiving recommended mental health first aid support was associated with perceived benefits by the recipient of help, including improved mental health (r = .37), a closer relationship with their helper (r = .39), and impact on help seeking from a doctor (r = .45), mental health professional (r = .49) or a phone or digital mental health service (r = .38). These findings provide empirical evidence for the potential benefits of upskilling community members in effective mental health first aid support.

## Introduction

Globally, mental disorders and substance use disorders affect about 15% of the community and are a key contributor to the burden of disease [1]. This high prevalence

**Data availability statement:** The data that support the findings of this study are available on Figshare https://doi.org/10.26188/30561206.v1.

**Funding:** This work was supported by a Mental Health First Aid International Fellowship (AJM), a Veski FAIR Fellowship (AJM), and a National Health and Medical Research Council Investigator Grant (1172889, AFJ). The funders had no role in study design, data collection and analysis, decision to publish, or preparation of the manuscript.

**Competing interests:** I have read the journal's policy and the authors of this manuscript have the following competing interests: AJM receives funding from Mental Health First Aid International (MHFAI) and is a member of MHFAI's International Evaluation and Research Advisory Committee. NJR Chairs MHFAI's International Evaluation and Research Advisory Committee. AFJ is a former member of the board of MHFAI. All other authors declare no conflicts of interest.

means many people will know someone with a mental health problem in their personal network. The role of friends and family members in providing mental health support can be complex. Discrimination from family and friends is a common experience reported by people with mental health problems, such as people avoiding or excluding them or expressing judgemental and dismissive attitudes about their problem or the need for treatment [2,3]. In contrast, friends and family members can also act as initial facilitators of help-seeking [4–6] by recognising emerging symptoms, deciding to intervene, and encouraging professional help, especially when it is considered appropriate to their relationship and they have the ability and capacity to help [7]. This is important because world-wide data show there remains a significant gap in the uptake of professional treatments for mental health problems [8] and there can be considerable delays before help is sought [9,10]. Friends and family members can also be important sources of support during treatment and recovery, by providing emotional support, practical help with everyday tasks and navigating the health system, and encouraging engagement in treatment and self-care activities [2,11,12]. Improving the knowledge and mental health support skills of family members and friends is therefore one avenue to facilitating early intervention and improving population mental health [13].

Over the last two decades there has been widespread implementation of community education about mental health problems in schools and the community. A prominent example is Mental Health First Aid (MHFA) training, which has been undertaken by more than 8 million people across 30 countries [14]. MHFA courses teach members of the public how to support someone who is developing a mental health problem or is in crisis, such as being suicidal or having a panic attack. As well as aiming to improve knowledge about mental health problems and evidence-based treatments, these courses teach practical skills in how to provide 'mental health first aid'. This is defined as the help that is provided until appropriate professional help is received or the crisis resolves [15]. The content of MHFA courses is based on guidelines, developed through expert consensus, on how to provide mental health first aid [16]. This revolves around the MHFA Action Plan: A (approach the person, assess and assist with any crisis); L (Listen and communicate non-judgementally); G (Give support and information - including emotional support, practical help, and information about mental health problems); E (Encourage appropriate professional help); and E (Encourage other supports, such as self-help strategies and support from other loved ones) [15]. As such, providing 'mental health first aid' overlaps with providing social support, as it includes types of help consistent with emotional support, informational support and tangible support, which are typically included in the construct of enacted or received social support [17]. But mental health first aid is distinguished by its specific context, purpose, and clarity in helpful versus unhelpful forms of support.

MHFA courses have been subject to extensive evaluation. Meta-analyses of controlled trials have shown that the program is effective in improving knowledge about mental health problems and their treatments, stigmatising attitudes, and confidence and intentions to provide mental health first aid in trainees [18,19]. As with any training course, it is more challenging to understand whether the skills taught

in the course are applied in the real world and have successful outcomes. There is limited evidence from randomised controlled trials on outcomes from applying MHFA skills, since participants are course trainees rather than those receiving aid [20,21]. In addition, only a subset of participants will have the opportunity to support someone during a trial follow-up, leading to a lack of power to detect outcomes [22]. Surveys of past trainees suggest the skills are utilised after training [23,24] and one survey reported 88% of participants perceived their actions were helpful for the recipient and that professional help had been sought in 65% of cases [25]. Qualitative studies that have explored experiences of providing mental health first aid also report positive outcomes in recipients, as well as benefits to the helper themselves, such as a sense of satisfaction from helping and a stronger relationship with the recipient [7,26,27]. Nevertheless, these studies explore the perspective of providing mental health first aid, and as such, do not directly evaluate the impact on the recipients of help. An alternative approach is to investigate the impact of receiving support that is consistent with best practice mental health first aid, regardless of who provided the support or what training they had. This approach focuses on the effects of the *skills* taught in the training, rather than evaluating the training course specifically.

The Mental Health Support Scale (MHSS) was developed to measure mental health first aid skills that have been provided to others [28], but no validated scale exists to measure mental health first aid support from the perspective of the recipient of help. The MHSS is a self-report measure of mental health first aid skills developed from best-practice guidelines on how to provide mental health first aid. Potential items were selected with item response theory to provide measurement precision across a range of skill levels and to cover all parts of the MHFA Action Plan. The scale discriminated between respondents with and without mental health first aid expertise and correlated with higher mental health first aid knowledge and mental health literacy and lower stigmatising attitudes [28]. It has since been culturally adapted and validated for use in China [29] and Chile and Argentina [30]. As MHSS items describe observable actions (e.g., "Asked them whether they had thoughts of harming themselves or others") it is possible to adapt the scale so that it could be completed by the person receiving help. This would allow for direct measurement of impacts from receiving different levels of support, such as 'high quality' support consistent with best practice mental health first aid, versus support that does not follow best practice.

This study's aims were twofold: (1) to adapt the Mental Health Support Scale to a version that could be completed by the recipient of help and evaluate its reliability and validity, and (2) to use this measure to investigate the outcomes from receiving mental health first aid, specifically whether the quality of mental health support received was associated with the perceived impact on mental health, relationship with the helper, and seeking professional help.

## Materials and methods

### Ethics statement

Participants provided online written consent and the project received ethics approval from the University of Melbourne Human Research Ethics Committee (24907).

### Participants

Participants were adults with recent experience of mental ill-health. Eligibility criteria included (1) an adult aged 18+; (2) in the past 12 months, has developed a mental health problem, experienced a worsening of an existing mental health problem, or had a mental health crisis (e.g., were suicidal); (3) received support for that problem from someone they know well, such as a family member, friend or colleague; (4) fluent in English; and (5) lives in a high income, English-speaking country (United States, United Kingdom, Australia, Canada).

Participants were recruited through Prolific between 14–29 December 2022. The study was advertised to participants who had answered yes to one of two prescreening questions that Prolific asks all registered users when they sign up: (1) *Do you have – or have you had – a diagnosed, on-going mental health/illness/condition?*; (2) *Do you have any diagnosed*

*mental health condition that is uncontrolled (by medication or intervention) and which has a significant impact on your daily life/ activities?*. In addition, in order to maximise the likelihood of quality responding, participants were limited to those who had an approval rate of 90% or above and had completed 50 or more studies on Prolific. We aimed to recruit 1100 participants, in order to achieve a minimum 200 participants for a planned 6-month follow-up study.

## Procedure

Questionnaires were hosted online using Qualtrics software. A brief screening survey first asked whether participants had developed a mental health problem, experienced a worsening of an existing mental health problem, or had a mental health crisis in the past 12 months, and whether they received support from someone they knew well. Those who screened positive were invited to participate, where they completed the measures described below. Two attention check questions were embedded in the questionnaire, to filter out data from participants who were not attentive. Participants who failed both attention check questions were rejected for payment and excluded from analysis. Participants received a payment equivalent to UK £8/hour for completing the surveys. Upon completion, participants were encouraged to contact their local doctor or mental health provider if the survey had brought up difficult emotions or they wanted mental health support and were also provided with a list of confidential support options they could access (e.g., Samaritans helpline in the UK).

## Measures

**Mental Health Support Scale-Received (MHSS-Received).** As described above, the Mental Health Support Scale (MHSS) is a self-report measure of mental health first aid skills applicable to a range of mental health problems [28]. It includes two factors of recommended and not recommended actions based on MHFA guidelines. The Provided version can be used when the respondent has provided mental health first aid to a person experiencing a mental health problem or crisis. We developed the MHSS-Received by adapting the items in the MHSS-Provided so that they were phrased from the perspective of receiving help rather than providing it. There are 12 core items that all respondents can answer with three options: 'Yes', 'No', or 'Not sure'. There are also 11 supplementary items relevant to when the person was suicidal, experiencing psychosis, or reluctant to seek help, which are presented when appropriate based on answers to screening questions. Items were dichotomised into 'correct' versus 'incorrect' (based on MHFA guidelines), such that responses of 'Yes' to recommended items were scored 1, responses of 'No' to not recommended items were scored 1, and all other responses were scored 0. Higher scores therefore indicate better quality support that is consistent with best practice.

**Multidimensional Evaluation of Enacted Social Support.** We used the Multidimensional Evaluation of Enacted Social Support (MEESS) [31] to evaluate the convergent validity of the MHSS-Received. The MEESS was designed to measure respondents' evaluations of social support they provide, receive, or observe. Respondents evaluate adjective pairs that reflect opposite ends of a continuum (e.g., helpful–unhelpful) and select a number between 1 and 7 to indicate their evaluation of the support for each criterion. These 12 semantic differential items represent three kinds of support: emotional awareness (e.g., sensitive-insensitive), relational assurance (e.g., encouraging-discouraging), and problem-solving utility (e.g., ignorant-knowledgeable). The MEESS has shown acceptable psychometric properties, with high internal consistency and evidence for predictive validity [32]. We used the mean score of the 12 items (omega = 0.96) [33]. We hypothesized that more positive evaluations of support, as indicated by higher scores on the MEESS, would be positively correlated with higher scores on the MHSS-Received.

**Multidimensional Scale of Perceived Social Support.** The Multidimensional Scale of Perceived Social Support (MSPSS) [34] measures perceptions of the availability of support from three sources including family, friends and a significant other. The MSPSS was used to assess evidence for divergent (or discriminant) validity, as the availability of social support is a related yet distinct construct to the quality of mental health first aid support [17]. We used the

6-item short form, which assesses the availability of support using a 7-point Likert scale. Items can be combined to generate subscale scores per support source or a total score (omega = 0.83 for the total score). We hypothesized that the correlation between the MSPSS and the MHSS would be smaller than the correlation between the MEESS and the MHSS.

**Perceived impact.** To assess the perceived impact of the support on the respondent's relationship with their helper, respondents were asked "How did the support you received from your [Family member/Intimate partner/Friend/Work colleague] affect your relationship with them?" with response options "5=Strengthened it a lot, 4=Strengthened it a little bit, 3=No change, 2=Weakened it a little, 1=Weakened it a lot". We hypothesised that higher scores would be associated with higher scores on the MHSS-Received.

To assess the perceived impact of the support on their mental health, respondents were asked "How did the support you received from your [Family member/Intimate partner/Friend/Work colleague] affect your mental health?" and "How has your mental health changed over the past 6 months?" with the response scale "Did it make it: 5 = Much better, 4 = A bit better, 3 = No change, 2 = A bit worse, 1 = Much worse". We hypothesised that higher scores on these questions would be associated with higher scores on the MHSS-Received. These items showed a medium-sized correlation (r = .35, [95% CI.30-.41]).

**Professional help.** Impact on seeking professional help was assessed with the Actual Help-Seeking Questionnaire [35]. This questionnaire lists a number of potential help sources and asks whether or not the respondent has sought help from each. We included four sources of professional help (GP or family doctor, mental health professional, mental health or crisis phone helpline, digital or telehealth mental health service) and six sources of informal help (partner, friend, parent, other relative, work colleague, religious leader). In addition, if respondents indicated they sought help from any of the professional sources, we asked about the role the helper played, e.g., "The support from my [Family member/ Intimate partner/Friend/Work colleague] helped me to seek help from a Doctor/GP; Strongly disagree, disagree, not sure, agree, strongly agree". We hypothesised that better mental health support would be positively associated with seeking professional help.

**Other questions.** Respondents completed the K6 measure of psychological distress [36] and identified their mental health problem, age, gender, highest level of education, language spoken at home, country, marital status, employment status, financial satisfaction and provided any feedback or comments. In addition, we asked about the respondent's relationship to their helper (e.g., family member, friend) and the helper's gender.

## Statistical analysis

To verify the dimensionality of the scales, a confirmatory factor analysis was conducted on MHSS-Received items to test the fit of a two-factor solution (recommended and not-recommended actions). As data were binary, we estimated model parameters using diagonally weighted least squares (DWLS) estimator, denoted 'weighted least squares mean and variance adjusted' (WLSMV) in lavaan (R). Model fit was evaluated against established cut-off values for excellent fit of ≥0.95 for Comparative Fit Index (CFI) and Tucker-Lewis Fit Index (TLI) and values ≤0.05 for Root Mean Square Error of Approximation (RMSEA), whereas acceptable model fit was determined using cut-off values of ≥0.90 for CFI and TLI and values ≤0.08 for RMSEA [37,38].

Pearson's or Spearman's correlation coefficients were calculated to examine convergent and divergent validity and associations with impact. Internal consistency was calculated using McDonald's Omega and the Spearman-Brown coefficient for two-item scales [39]. The size of correlation coefficients was interpreted as r = 0.1 (small), r = 0.3 (medium) and r = 0.5 (large). Correlates of mental health support (recipient gender, helper gender, type of recipient-helper relationship, mental health problem, age, education, language spoken at home) were explored with a multiple linear regression model. CFA was performed using the lavaan package (0.6-19) in R and all other analyses were conducted in Stata 18.

## Results

We received 1156 responses, of which 39 did not complete the questionnaire, and 1 was excluded as their age was given as 17. Only six participants failed one of two attention check questions, leaving 1116 participants. Participant characteristics are shown in Table 1. About two-thirds were female, 52% reported a tertiary education, and a quarter reported being unemployed or unable to work. Participants reported a range of mental health problems, with depression and anxiety being most common. Psychological distress levels were moderate on average (M = 11.22, SD = 5.20). The person who helped them was most commonly an intimate partner (44.1%), family member (28.9%), friend (24.4%), work colleague (2.2%) and other (0.4%). The helper's gender was female (55.2%), male (43.2%) and other (1.6%).

Items on the MHSS-Received varied in frequency, with the most common being "Listened to your problems and tried to solve them" (a not-recommended action, see Table 2). The least common was "Asked whether you had thoughts of harming yourself or others". For the supplementary items, 517 (46.3%) reported having had thoughts of suicide, 124 (11.1%) had been at immediate risk of suicide, 151 (13.5%) had experienced psychosis, and 688 (61.6%) had been reluctant to seek professional help.

### MHSS-Received psychometrics

**Structural validity.** A CFA specifying two factors of recommended vs not recommended actions did not show acceptable fit ($\chi^2$ = 350.03, df = 54, p < .0001, RMSEA = 0.07, 90% CI: 0.06-0.08; CFI = 0.85; TLI = 0.82). In addition, the parameter estimates for the not-recommended items had relatively large standard errors and were not statistically significant. Inspection of these three items (#3, #6, #8) showed that item 3 had an extreme endorsement pattern and low correlations with the other two items (r = .13 and .15). Removing this item and correlating the residuals between items 4 and 5 (recommended actions which are similar in content and cover how to communicate, including repeating or restating things), indicated acceptable model fit ($\chi^2$ = 134.61, df = 42, p < .0001, RMSEA = 0.05, 90% CI: 0.04-0.06; CFI = 0.93; TLI = 0.91).

The MHSS-R Recommended Actions subscale was formed by summing the nine recommended items, which had excellent reliability (omega = 0.81). On average, participants reported receiving support that was at the midpoint of the scale (see Table 3 for means and standard deviations for all study outcomes). The Not Recommended Actions subscale was formed by summing the two not recommended actions (reverse scored), such that higher scores indicate better quality support ($\rho$ = .35).

Total scores were calculated for each supplementary set of items. The immediate risk of suicide items had acceptable reliability (omega = 0.71). The two items on the reluctant to seek help subscale were combined ($\rho$ = .62). The suicide and psychosis sets of items contained one not-recommended item each. These items loaded negatively on each factor, despite reverse-scoring them so that all items were keyed in the same direction. As such, the suicide and psychosis items were not combined to form a unidimensional scale.

**Convergent and divergent validity.** To examine convergent validity, the correlation between scores on the MEESS and MHSS-R were calculated. As hypothesised, there was a positive correlation between MEESS scores and both MHSS-R Recommended Actions scores (r = .41 [.36,.46]) and MHSS-R Not Recommended Actions scores (r = .24 [.19,.30]) (see Table 4).

There were also small to medium correlations between MSPSS scores and scores on the MHSS-R Recommended Actions subscale (r = .28 [.23,.33]) and Not Recommended Actions subscale (r = .14 [.09,.20]), which were smaller than the correlations with the MEESS, providing evidence for divergent validity.

### Quality of received help received and impact on recipients

**Impact on mental health and relationship.** Results on the perceived impact of received support are shown in Table 3. A majority of participants reported that the support they received from their helper strengthened their relationship with

**Table 1. Participant demographics.**

| Characteristic | Total (n = 1116) |
|---|---|
| Age, M (SD) range | 34.4 (11.4) 18-76 |
| Gender, n (%) | |
| Female | 716 (64.2) |
| Male | 342 (30.7) |
| Other[a] | 58 (5.2) |
| Marital status, n (%) | |
| Never married | 556 (49.8) |
| Married, de facto | 476 (42.7) |
| Separated, divorced or widowed | 84 (7.5) |
| Highest level of education, n (%) | |
| Primary/elementary school | 9 (0.8) |
| Secondary/high school | 325 (29.1) |
| Certificate, trade or apprenticeship | 190 (17.0) |
| Bachelor degree | 423 (37.9) |
| Postgraduate degree | 157 (14.1) |
| Other | 12 (1.1) |
| Speaks a language other than English at home, n (%) | 165 (14.8) |
| Employment status, n (%) | |
| Full-time | 533 (47.8) |
| Part-time | 195 (17.5) |
| Unemployed | 148 (13.3) |
| Unable to work | 122 (10.9) |
| Contract/temporary/casual | 54 (4.8) |
| Other | 60 (5.4) |
| Prefer not to say | 4 (0.4) |
| Country of residence, n (%) | |
| United Kingdom | 463 (41.5) |
| United States | 304 (27.2) |
| Canada | 248 (22.2) |
| Australia | 101 (9.1) |
| Financial satisfaction, M (SD) range | 4.9 (2.3) 1-10 |
| Self-reported mental health problem or crisis, n (%) | |
| Depression | 855 (76.6) |
| Attempted suicide or self-harm | 121 (10.8) |
| Anxiety or anxiety disorder | 835 (74.8) |
| Agoraphobia or panic disorder | 121 (10.8) |
| Social phobia | 189 (16.9) |
| Generalised anxiety disorder | 262 (23.5) |
| (Any anxiety disorder) | 903 (80.9) |
| PTSD | 213 (19.1) |
| OCD | 113 (10.1) |
| Eating disorder/anorexia/bulimia | 108 (9.7) |
| Schizophrenia/paranoid schizophrenia | 4 (0.4) |
| Schizoaffective disorder | 2 (0.2) |
| Psychosis | 14 (1.3) |

*(Continued)*

**Table 1.** (Continued)

| Characteristic | Total (n = 1116) |
|---|---|
| Bipolar disorder | 72 (6.5) |
| Personality disorder | 63 (5.7) |
| ADHD | 166 (14.9) |
| Alcohol problem | 81 (7.3) |
| Drug addiction | 44 (3.9) |
| Something else | 33 (3.0) |

[a]Transgender, non-binary or gender diverse, other.

**Table 2. Mental Health Support Scale – Received item level responses**[a.]

| Item | Yes % | No % | Not sure % | Correct[b] % | Incorrect % |
|---|---|---|---|---|---|
| 1. Asked whether you had thoughts of harming yourself or others | 34.1 | 61.1 | 4.8 | 34.1 | 66.0 |
| 2. Discussed with you your wishes about privacy and confidentiality | 34.3 | 56.9 | 8.8 | 34.3 | 65.7 |
| 3. **Listened to your problems and tried to solve them | 91.5 | 4.6 | 3.9 | 4.6 | 95.4 |
| 4. Let you know they were listening to what you were saying by restating and summarising what you had said | 66.4 | 17.4 | 16.2 | 66.4 | 33.6 |
| 5. Communicated clearly and simply, and repeated things where necessary | 78.8 | 9.5 | 11.7 | 78.8 | 21.2 |
| 6. **Told you that you had to get your act together | 13.5 | 81.7 | 4.8 | 81.7 | 18.3 |
| 7. Conveyed a message of hope by telling you that help is available and things can get better | 79.1 | 10.8 | 10.0 | 79.1 | 20.9 |
| 8. **Tried to cheer you up by telling you that things don't seem that bad | 36.1 | 54.1 | 9.8 | 54.1 | 45.9 |
| 9. Offered you information and resources appropriate to your situation | 42.7 | 44.5 | 12.7 | 42.7 | 57.3 |
| 10. Discussed with you your options for seeking professional help | 58.7 | 33.9 | 7.4 | 58.7 | 41.3 |
| 11. Asked whether you had other supportive people you could rely on | 40.8 | 49.6 | 9.7 | 40.8 | 59.2 |
| 12. Discussed with you whether you were interested in self-help strategies | 43.9 | 43.4 | 12.7 | 43.9 | 56.1 |
| Supplementary items | | | | | |
| Had thoughts of suicide (n = 517) | | | | | |
| 1. Asked if you had been thinking about suicide | 47.2 | 46.6 | 6.2 | 47.2 | 52.8 |
| 2. **Told you how much it would hurt your family and friends if you were to kill yourself | 44.1 | 48.6 | 7.4 | 48.6 | 51.5 |
| 3. **Tried to make you understand that suicide is wrong | 29.8 | 58.8 | 11.4 | 58.8 | 41.2 |
| Immediate risk of suicide (n = 124) | | | | | |
| 1. Asked if you had a plan for suicide – for example, how, when and whether you intend to die | 40.3 | 53.2 | 6.5 | 40.3 | 59.7 |
| 2. Encouraged you to get appropriate professional help as soon as possible – for example, see a mental health professional or someone at a mental health service | 77.4 | 14.5 | 8.1 | 77.4 | 22.6 |
| 3. Made sure you were not left on your own | 66.9 | 28.2 | 4.8 | 66.9 | 33.1 |
| Experienced psychosis (n = 151) | | | | | |
| 1. Acknowledged that you might be frightened by what you were experiencing | 68.2 | 19.2 | 12.6 | 68.2 | 31.8 |
| 2. **Tried to convince you that your beliefs and perceptions were false | 38.4 | 47.7 | 13.9 | 47.7 | 52.3 |
| 3. Listened to you talk about your experiences even though they were not based in reality | 78.8 | 11.3 | 9.9 | 78.8 | 21.2 |
| Reluctant to seek help (n = 688) | | | | | |
| 1. Found out if there were specific reasons why you did not want to seek professional help | 49.9 | 39.2 | 10.9 | 49.9 | 50.2 |
| 2. Let you know you could contact them if you changed your mind about seeking help | 61.2 | 28.2 | 10.6 | 61.2 | 38.8 |

[a]** indicates a not recommended action.

[b]Correct if responded Yes to recommended actions, and No to not recommended actions.

 

**Table 3.  Means and standard deviations for outcome measures.**

| Measure/question | M (SD)/ n (%) | Possible range |
|---|---|---|
| MHSS Received Recommended Actions subscale, M (SD) | 4.79 (2.25) | 0-9 |
| MHSS Received Not Recommended Actions subscale, M (SD) | 1.36 (0.69) | 0-2 |
| MHSS Received (Immediate risk of suicide items), M (SD) | 1.85 (0.99) | 0-3 |
| MHSS Received (Reluctant to seek help items), M (SD) | 1.11 (0.84) | 0-2 |
| Multidimensional Evaluation of Enacted Social Support, M (SD) | 5.78 (1.03) | 1-7 |
| Multidimensional Scale of Perceived Social Support, M (SD) | 5.00 (1.21) | 1-7 |
| Perceived effect of support on relationship with helper, M (SD) | 4.03 (0.89) | 1-5 |
| Strengthened it a lot, n (%) | 400 (45.8) | |
| Strengthened it a little bit, n (%) | 406 (36.4) | |
| No change, n (%) | 265 (23.8) | |
| Weakened it a little, n (%) | 38 (3.4) | |
| Weakened it a lot, n (%) | 7 (0.6) | |
| Perceived effect of support on mental health, M (SD) | 3.99 (0.65) | 1-5 |
| Much better, n (%) | 189 (16.9) | |
| A bit better, n (%) | 758 (67.9) | |
| No change, n (%) | 140 (12.5) | |
| A bit worse, n (%) | 24 (2.1) | |
| Much worse, n (%) | 5 (0.5) | |
| Perceived impact of support on seeking help from GP, M (SD)[a] | 3.67 (1.07) | 1-5 |
| Strongly disagree, n (%) | 12 (3.2) | |
| Disagree, n (%) | 61 (16.1) | |
| Not sure, n (%) | 45 (11.9) | |
| Agree, n (%) | 183 (48.3) | |
| Strongly agree, n (%) | 78 (20.6) | |
| Perceived impact of support on seeking help from mental health professional, M (SD)[b] | 3.75 (1.08) | 1-5 |
| Strongly disagree, n (%) | 13 (2.6) | |
| Disagree, n (%) | 79 (16.0) | |
| Not sure, n (%) | 52 (10.5) | |
| Agree, n (%) | 226 (45.8) | |
| Strongly agree, n (%) | 124 (25.1) | |
| Perceived impact of support on seeking help from phone helpline, digital or telehealth mental health service, M (SD)[c] | 3.32 (1.22) | 1-5 |
| Strongly disagree, n (%) | 11 (6.1) | |
| Disagree, n (%) | 48 (26.7) | |
| Not sure, n (%) | 27 (15.0) | |
| Agree, n (%) | 61 (33.9) | |
| Strongly agree, n (%) | 33 (18.3) | |
| Change in mental health in past 6 months, M (SD) | 3.68 (1.14) | 1-5 |
| Much better, n (%) | 266 (23.8) | |
| A bit better, n (%) | 514 (46.1) | |
| No change, n (%) | 104 (9.3) | |
| A bit worse, n (%) | 181 (16.2) | |
| Much worse, n (%) | 51 (4.6) | |

[a] N = 379.

[b] N = 494.

[c] N = 180.

**Table 4. Correlations between Mental Health Support Scale – Received and other measures.**

| | MEESS | MSPSS | Impact on helper relationship | Impact on mental health | Change in mental health in past 6 months | Help-seeking (GP/doctor) | Help-seeking (Mental health professional) | Help-seeking (phone/digital) |
|---|---|---|---|---|---|---|---|---|
| MHSS-R: Recommended Actions | 0.41*** | 0.28*** | 0.39*** | 0.37*** | 0.21*** | 0.45*** | 0.49*** | 0.38*** |
| MHSS-R: Not recommended Actions | 0.24*** | 0.14*** | 0.17*** | 0.16*** | 0.04 | 0.01 | -0.08 | -0.14 |
| MHSS-R Suicide-1: Asked if you had been thinking about suicide | 0.22*** | 0.13** | 0.20*** | 0.17*** | 0.10* | 0.20** | 0.32*** | 0.16 |
| MHSS-R Suicide-2: Told you how much it would hurt your family and friends if you were to kill yourself | -0.13** | -0.15*** | -0.11* | -0.11* | -0.12** | -0.17* | -0.24*** | -0.10 |
| MHSS-R Suicide-3: Tried to make you understand that suicide is wrong | -0.15** | -0.16*** | -0.12** | -0.11* | -0.14** | -0.10 | -0.23*** | -0.16 |
| MHSS-R Immediate risk of suicide | 0.35*** | 0.38*** | 0.17 | 0.24** | 0.06 | 0.61*** | 0.58*** | 0.46** |
| MHSS-R Psychosis-1: Acknowledged that you might be frightened by what you were experiencing | 0.33*** | 0.27*** | 0.33*** | 0.30*** | 0.26** | 0.06 | -0.03 | 0.00 |
| MHSS-R Psychosis-2: Tried to convince you that your beliefs and perceptions were false | 0.06 | -0.13 | -0.13 | -0.13 | -0.10 | 0.08 | -0.23* | -0.23 |
| MHSS-R Psychosis-3: Listened to you talk about your experiences even though they were not based in reality | 0.19* | 0.18* | .08 | 0.17 | 0.30*** | 0.20 | 0.04 | -0.17 |
| MHSS-R Reluctance to seek help | 0.30*** | 0.27*** | 0.36*** | 0.32*** | 0.15*** | 0.37*** | 0.37*** | 0.34*** |

Note: * $p < .05$, ** $p < .01$, *** $p < .001$.

them and improved their mental health. As shown in Table 4, there were medium-to-large positive correlations between scores on the MHSS-R Recommended Actions subscale and the perceived impact on the relationship with the helper and the perceived impact on participant mental health. There was also a small-to-medium correlation with change in mental health in the past 6 months. Correlations between scores on the MHSS-R Not Recommended Actions subscale and impact on helper relationship and mental health were also small but in the hypothesised direction.

For the supplementary items, we observed generally small-to-medium sized correlations, although the not-recommended suicide items were in the negative direction.

**Impact on help-seeking.** There were 379 (34.0%) participants who had sought help from a GP or doctor, 494 (44.3%) from a mental health professional, and 180 (16.1%) from a phone helpline, digital or telehealth mental health service. As shown in Table 4, there were large positive correlations between scores on the Recommended Actions subscale and perceived impact of their helper's support on seeking help from a GP/doctor and from a mental health professional. There was also a medium-to-large positive correlation with seeking help from a phone or digital mental health service. In contrast, scores on the Not Recommended Actions subscale showed no correlation with seeking professional help.

The associations with help-seeking were large for the immediate risk of suicide scale and medium sized for the reluctance to seek help scale. Associations were mixed for the suicide items, depending on whether they were recommended or not recommended. There were few significant correlations observed for the psychosis items, but the sample size was much smaller (e.g., 59 participants with psychosis had sought help from their GP).

## Correlates of receiving quality support

Table 5 presents the correlates of receiving quality support, based on the MHSS-R Recommended Actions subscale and Not Recommended Actions subscale. Across both scales, receiving lower quality support was associated with male

**Table 5. Multiple regression models predicting quality support.**

| Predictor | Recommended actions | | | Not recommended actions | | |
|---|---|---|---|---|---|---|
| | b[a] | 95% CI | p | b[a] | 95% CI | p |
| Gender of helper | | | | | | |
| Female helper (reference) | | | | | | |
| Male helper | -0.50 | [-0.80, -0.19] | **.001** | -0.16 | [-0.25, -0.06] | **.001** |
| Other helper | 0.31 | [-0.81, 1.43] | .589 | 0.27 | [-0.07, 0.62] | .114 |
| Gender of recipient | | | | | | |
| Female recipient (reference) | | | | | | |
| Male recipient | 0.36 | [0.06, 0.67] | **.018** | -0.19 | [-0.28, -0.10] | **<.001** |
| Other recipient | -0.23 | [-0. 86, 0.40] | .474 | 0.02 | [-0.17, 0.21] | .843 |
| Age of recipient | -0.02 | [-0.03, -0.00] | **.016** | -0.00 | [-0.00, 0.00] | .635 |
| Highest education level of recipient | | | | | | |
| Below tertiary level (reference) | | | | | | |
| Tertiary level | -0.10 | [-0.37, 0.17] | .458 | -0.05 | [-0.13, 0.03] | .219 |
| Language spoken at home of recipient | | | | | | |
| English only (reference) | | | | | | |
| Other | 0.33 | [-0.05, 0.70] | .087 | -0.15 | [-0.26, -0.03] | **.012** |
| Helper relationship | | | | | | |
| Intimate partner (reference) | | | | | | |
| Family member | -0.46 | [-0.80, -0.11] | **.009** | -0.20 | [-0.30, -0.10] | **<.001** |
| Friend | 0.03 | [-0.32, 0.37] | .881 | 0.11 | [0.00, 0.21] | **.047** |
| Work colleague | 0.99 | [0.08, 1.89] | **.033** | 0.09 | [-0.19, 0.36] | .527 |
| Other | 1.30 | [-0.51, 3.11] | .159 | -0.37 | [-0.92, 0.18] | .184 |
| Type of mental health problem | | | | | | |
| Depression | 0.25 | [-0.06, 0.56] | .118 | -0.06 | [-0.15, 0.04] | .251 |
| Attempted suicide or self-harm | 0.50 | [0.05, 0.95] | **.029** | -0.07 | [-0.20, 0.07] | .335 |
| Anxiety disorder | -0.04 | [-0.37, 0.30] | .833 | 0.03 | [-0.07, 0.13] | .554 |
| PTSD | 0.19 | [-0.15, 0.54] | .265 | 0.08 | [-0.02, 0.19] | .114 |
| OCD | 0.16 | [-0.28, 0.60] | .484 | -0.07 | [-0.21, 0.06] | .278 |
| Eating disorder | 0.32 | [-0.14, 0.78] | .176 | -0.06 | [-0.20, 0.08] | .371 |
| Severe mental illness | 0.32 | [-0.10, 0.73] | .133 | -0.01 | [-0.13, 0.12] | .929 |
| ADHD | -0.03 | [-0.41, 0.35] | .133 | -0.10 | [-0.22, 0.01] | .077 |
| Substance use disorder | -0.24 | [-0.69, 0.21] | .302 | -0.21 | [-0.34, -0.07] | **.003** |

[a]Unstandardised regression coefficient, higher scores indicate better quality support.

helpers and family member helpers, compared with intimate partners. We also explored whether there was an interaction between helper gender and recipient gender, but this was not significant for either Recommended Actions (p = .143) or Not Recommended Actions (p = .930).

## Discussion

This study aimed to evaluate an instrument measuring receipt of mental health first aid and explore the perceived impacts of receiving good quality mental health support. The Recommended Actions subscale of the Mental Health Support Scale-Received showed evidence for structural validity, internal consistency, and convergent and divergent validity. Receiving higher quality mental health support (consistent with best practice mental health first aid) was associated with

the perception of improved mental health, a strengthened relationship with the helper, and a greater impact on seeking help from a GP/doctor, mental health professional, or a phone or digital mental health service. The Not Recommended Actions subscale was also associated with mental health and relationship outcomes. These findings provide broad support for the perceived benefits of receiving support consistent with best-practice mental health first aid.

These findings underscore the important role of family and friends as community-based support networks who can contribute to increased early intervention, greater professional help-seeking, reduced stigma and discrimination and ultimately improved population mental health. Supportive, non-dismissive interactions are perceived as beneficial for recipient mental health, and encouragement to seek professional help does have an impact, consistent with prior research that has examined this issue [4–6,40]. Research within social psychology has shown that perceived social support (its quality and availability) is more strongly associated with health and wellbeing than the quantity of received support, and that the two are only moderately correlated [41]. Greater perceived social support is also associated with better outcomes in those with mental health problems such as depression [42]. Our study's findings are consistent with this body of research – it is not just the quantity of support that is influential on wellbeing and mental health but its quality, and some forms of support are not associated with better outcomes.

The findings from this study indicate that there remains a need to upskill community members in how to provide effective support, through courses such as MHFA training and gatekeeper training for suicide prevention. The average score of the Recommended Actions subscale was less than 5 on a 9-point scale, and action related to suicide prevention was particularly poor. Less than half of respondents at risk of suicide reported being asked if they were thinking about suicide, and for those who were at immediate risk of suicide, only two-fifths were asked whether they had a plan for suicide. While it is possible that respondents did not clearly indicate suicide risk to their helper, these findings are consistent with those from other studies showing that myths about suicide are prevalent in the community, including that asking a person about suicide could make them start thinking about it [43]. MHFA training aims to teach which techniques to use to provide help, and training participants report improved confidence in providing mental health first aid [19]. Equipping community members with the skills and the confidence to implement them is likely to lead to better quality support, given that uncertainty about what to say or do to help a person with a mental health problem is associated with poorer quality support [44].

This study has some limitations that should be acknowledged. The impacts on recipient mental health and relationship with their helper were perceived impacts assessed by single items that had not been previously validated. Other methods may be less biased, such as assessing mental health and relationship closeness longitudinally, before and after receiving support, with validated scales. Although this method would be a more objective measure of change, there could be several explanatory factors in any change, beyond the mental health support from one individual. Another limitation is that we did not enquire about how long ago the support was provided within the past 12 months, or over what period. The 12-month recall period may have introduced memory inaccuracies and increased measurement error. Despite this, most items had less than ten percent 'not sure' answers, suggesting that most participants could recall the interactions reasonably well.

Future research should focus on strengthening the not recommended actions of the Mental Health Support Scale - Received. One not recommended item, *Listened to your problems and tried to solve them,* did not correlate highly with the other items and was dropped, leaving only two items on the Not Recommended Actions subscale. It is noteworthy that this item was not included in cultural adaptations of the MHSS for China and Chile/Argentina because it did not perform well [29]. Further refinement of this item or its replacement should be considered. Although the Not Recommended Actions subscale showed evidence of convergent and divergent validity, adding more items would strengthen its reliability and ensure other aspects of not recommended support are measured. There were several candidate items from the development of the Provided version of the Mental Health Support Scale that could be tested in future research [28]. Furthermore, the not recommended actions to support a person thinking about suicide were unexpectedly associated with benefits. These actions *Told you how much it would hurt your family and friends if you were to kill yourself* and *Tried to make you understand that suicide is wrong* were based on expert consensus on what to avoid when supporting a suicidal person

[45]. A potential interpretation of our findings is that any discussion of suicidal thoughts, even if not optimal, is better than no discussion at all.

In addition, although this study's findings provide some empirical evidence to support MHFA guidelines developed through expert consensus, future work should examine whether MHFA *training* leads to improvements in mental health first aid skills, as measured by the Mental Health Support Scale. This is an important component in the program logic between training and impacts on recipients. If there is evidence that MHFA training leads to higher quality mental health support skills, and evidence that receiving high quality support leads to recipient benefits, then this is indirect evidence of the effectiveness of MHFA training.

In conclusion, the Recommended Actions subscale of the Mental Health Support Scale – Received version shows preliminary acceptable psychometric properties in measuring receipt of mental health first aid behaviours. Support that was consistent with best practice mental health first aid was associated with perceived benefits by the recipient of help, including improved mental health, a closer relationship with their helper, and impact on professional help seeking. These findings provide empirical evidence for the potential benefits of upskilling community members in effective mental health first aid support. Improving the mental health support provided within social networks could have a significant positive impact on population mental health.

## Author contributions

**Conceptualization:** Amy J Morgan, Nicola J Reavley, Anthony F Jorm.

**Data curation:** Amy J Morgan.

**Formal analysis:** Amy J Morgan, Andrew J Mackinnon.

**Funding acquisition:** Amy J Morgan, Anthony F Jorm.

**Investigation:** Judith Wright.

**Methodology:** Amy J Morgan, Judith Wright, Andrew J Mackinnon, Nicola J Reavley, Alyssia Rossetto, Long Khanh-Dao Le, Anthony F Jorm.

**Project administration:** Judith Wright.

**Writing – original draft:** Amy J Morgan.

**Writing – review & editing:** Judith Wright, Andrew J Mackinnon, Nicola J Reavley, Alyssia Rossetto, Long Khanh-Dao Le, Anthony F Jorm.

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
