## [Decision Letter · Decision Letter 0]

4 Nov 2025

PMEN-D-25-00367

Receiving support for mental health problems from family and friends: Measurement and impact on mental health, relationships, and help-seeking

PLOS Mental Health

Dear Dr. Morgan,

Thank you for submitting your manuscript to PLOS Mental Health. After careful consideration, we feel that it has merit but does not fully meet PLOS Mental Health’s publication criteria as it currently stands. Therefore, we invite you to submit a revised version of the manuscript that addresses the points raised during the review process.

We look forward to receiving your revised manuscript.

Kind regards,

Ang Li

Academic Editor

PLOS Mental Health

Journal Requirements:

1. Please describe in your methods section how capacity to provide consent was determined for the participants in this study. Please also state whether your ethics committee or IRB approved this consent procedure. If you did not assess capacity to consent please briefly outline why this was not necessary in this case.

i. State the initials, alongside each funding source, of each author to receive each grant.

ii. State what role the funders took in the study. If the funders had no role in your study, please state: “The funders had no role in study design, data collection and analysis, decision to publish, or preparation of the manuscript.”

3. Please send a completed 'Competing Interests' statement, including any COIs declared by your co-authors. If you have no competing interests to declare, please state "The authors have declared that no competing interests exist". Otherwise please declare all competing interests beginning with the statement "I have read the journal's policy and the authors of this manuscript have the following competing interests:"

4. Please ensure that your Ethics Statement is available in its entirety at the beginning of your Methods section, under a subheading 'Ethics Statement'.

5. In the online submission form, you indicated that “The data that support the findings of this study are available on request from the corresponding author, Amy Morgan.”.

3. Uploaded as supplementary information.

Additional Editor Comments (if provided):

Reviewer #2:

Reviewers' comments:

Reviewer's Responses to Questions

**Comments to the Author**

1. Does this manuscript meet PLOS Mental Health’s publication criteria?

Reviewer #1: Yes

Reviewer #2: Yes

Reviewer #3: Yes

2. Has the statistical analysis been performed appropriately and rigorously?

Reviewer #1: Yes

Reviewer #2: Yes

Reviewer #3: I don't know

3. Have the authors made all data underlying the findings in their manuscript fully available (please refer to the Data Availability Statement at the start of the manuscript PDF file)?

Reviewer #1: No

Reviewer #2: Yes

Reviewer #3: Yes

4. Is the manuscript presented in an intelligible fashion and written in standard English?

Reviewer #1: Yes

Reviewer #2: Yes

Reviewer #3: Yes

Reviewer #1: This review was removed by journal staff as the comments provided were not relevant to this submission. 

Reviewer #2: This manuscript presents a well-executed and timely investigation into the quality of informal mental health support and its impact on recipients. The research is methodologically sound, with a large sample size and robust psychometric validation, and contributes meaningfully to the literature on community-based mental health support.

Areas for Improvement

1) Recall Bias: The 12-month recall period may introduce memory inaccuracies; a shorter timeframe could improve reliability. Propably it has to be more clear indicated in the Limitation section

2) MHSS-Received psychometrics analysis distract from the main article`s content. Maybe it is possible at least the tables to move to the Annexes. Otherwise, I would recommend to publish these data as a separate article.

Reviewer #3: Thanks for the opportunity to review this interesting study on Mental Health First Aid and the development of a questionnaire for people who have received MHFA to self report from their perspective of receiving help. This kind of measure contribute towards the evaluation of such programmes and training for the individual receiving it on what is a very difficult to measure interpersonal process - when someone will use the training and in what way and how long for can be unpredictable and so this design focuses on the recipient to retrospectively consider interactions that resulted in them feeling helped, which can theoretically confirm that skills taught on MHFA are helpful.

Introduction

The introduction effectively outlines the key concepts related to MHFA and is clear and well written. There majority of the discussion is on MHFA and little on the key dynamics of the people being studied. This section could benefit from increased discussion on the role of family and friends in mental health support and in the help-seeking process. This area is often assumed or briefly mentioned despite being very complex. In acknowledgement of this, expanding briefly/signposting to other research on how informal sources of help act as helpers, blockers, facilitators to help, support while getting help, maintenance, out of hours support etc. can deepen the understanding on why MHFA is so important for communities, with ripple effects into society on stigma, discrimination, or misunderstanding of mental health issues, and larger impacts on health/life outcomes. Expanding this discussion would both enhance the literature linkage and provide stronger justification for the research focus. This links with the discussion.

Methodology

The description of participants, sampling procedures, and measures is clear and appropriately detailed. The adaptation of the MHSS to the (MHSS-Received) appears transparent and the use of a wide range of established measures for validity checking is thoroughly documented. One area that could be improved is further information on debriefing and/or aftercare. Being transparent about these processes is important for information sharing and best practice reporting. Including a brief discussion of how participants were supported post-study (e.g., debriefing procedures or access to resources) would demonstrate ethical rigor and good research practice.

Results and discussion

The results are well-presented, confirming what is known or what is being taught on MHFA courses. There are clear tables and effective use of narrative summaries. The discussions on validity and interpretation are well-aligned with the data. The discussion makes a persuasive case for the benefits of mental health first aid training. However, it could go further in articulating why these findings matter beyond the immediate context. This links back with the comments in the introduction, authors could emphasize how upskilling community members in mental health first aid could contribute to broader public health outcomes -such as increasing early intervention, reducing stigma, and strengthening community-based support networks. Expanding on these implications would increase the practical relevance and societal value of the study.

**Do you want your identity to be public for this peer review?** For information about this choice, including consent withdrawal, please see our Privacy Policy

Reviewer #1: No

Reviewer #2: **Yes: ** Vitalii Klymchuk

Reviewer #3: No

---

## [Editor Report · Decision Letter 1]

18 Nov 2025

Receiving support for mental health problems from family and friends: Measurement and impact on mental health, relationships, and help-seeking

PMEN-D-25-00367R1

Dear Dr Morgan,

We are pleased to inform you that your manuscript 'Receiving support for mental health problems from family and friends: Measurement and impact on mental health, relationships, and help-seeking' has been provisionally accepted for publication in PLOS Mental Health.

Best regards,

Ang Li

Academic Editor

PLOS Mental Health